# Reversible Transformations of Palladium–Indium Intermetallic Nanoparticles upon Repetitive Redox Treatments in H₂/O₂



Andrey V. Bukhtiyarov [1], Maxim A. Panafidin [1], Igor P. Prosvirin [1], Nadezhda S. Smirnova [1,2], Pavel V. Markov [1,2], Galina N. Baeva [2], Igor S. Mashkovsky [2], Galina O. Bragina [2], Zakhar S. Vinokurov [1], Yan V. Zubavichus [1,*], Valerii I. Bukhtiyarov [1] and Alexander Yu. Stakheev [2]

[1] Boreskov Institute of Catalysis, Lavrentieva Ave. 5, 630090 Novosibirsk, Russia; avb@catalysis.ru (A.V.B.); mpanafidin@catalysis.ru (M.A.P.); prosvirin@catalysis.ru (I.P.P.); felisine@gmail.com (N.S.S.); markovp@mail.ru (P.V.M.); vinokurovzs@catalysis.ru (Z.S.V.); vib@catalysis.ru (V.I.B.)

[2] Zelinsky Institute of Organic Chemistry, Leninsky Prospect 47, 119991 Moscow, Russia; bg305@ioc.ac.ru (G.N.B.); im@ioc.ac.ru (I.S.M.); bragina@ioc.ac.ru (G.O.B.); st@ioc.ac.ru (A.Y.S.)

* Correspondence: yvz@catalysis.ru; Tel.: +7-383-326-97-87

**Abstract:** The transformations of chemical states and structures occurring in the PdIn/Al₂O₃ catalyst upon redox treatments in different gaseous atmospheres at different temperatures are addressed by an assortment of in situ bulk- (XRD) and surface-sensitive (XPS and DRIFTS CO) techniques. Any desired state of the catalyst between two opposite extremes of highly dispersed oxide species and regularly ordered PdIn intermetallic compound could be set in fully controlled and reversible ways by selecting appropriate conditions for the reductive treatment starting from the fully oxidized state. Since mutual conversions of multi-atomic $Pd_n$ centers into single-site $Pd_1$ centers are involved in these transformations, the methodology could be used to find an optimum balance between the activity and selectivity of the catalytic system.

**Keywords:** PdIn intermetallic compound; selective acetylene hydrogenation; redox treatments; XRD; XPS; DRIFTS CO

## 1. Introduction

Bimetallic systems are widely used in heterogeneous catalysts due to their ability to enhance catalytic properties compared to monometallic analogs [1–14]. Despite the large number of publications devoted to investigations into the origin of such synergistic effects in bimetallic catalysts, they are still not fully understood. The majority of researchers agree that it is the structure and composition of the surface of a multicomponent catalytic system that serve as the key factors determining its catalytic properties [14–23]. It is known that a target-oriented regulation of catalytic properties can be achieved by various catalyst surface treatments, including the usage of the adsorbate-induced segregation effects [24]. The application of CO-induced segregation for the targeted tuning of active sites on the surface of Pd-based substitutional solid solutions, such as Pd-Cu [25,26], Pd-Ag [27–29] or Pd-Au [30–34] alloys, is well known. From the other side, it is also known that intermetallic catalytic systems (IMCs), for example, Pd-In [35] or Pd-Ga [36,37], are more stable against such segregation effects due to their higher thermodynamic stability compared to substitutional alloys [38,39]. An oxidative treatment of the surface of intermetallic compounds promotes the preferential oxidation of one of the components, which is more chemically active («oxidative leaching»). In the case of PdIn IMCs, such a process is accompanied by the segregation of indium from the bulk to the surface of PdIn nanoparticles, resulting in the formation of an In-depleted intermetallic phase [40]. According to our previously published works, this surface transformation could be used to obtain an optimum balance between the activity and selectivity of the catalyst in the selective C≡C to C=C bond hydrogenation reactions [41,42]. For example, in [42], the influence of the oxidative and reductive

treatments on the surface structure of PdIn nanoparticles was studied as a means for the deliberate tuning of their catalytic performance in a liquid-phase diphenylacetylene hydrogenation reaction using the combination of diffuse reflectance infrared Fourier-transform spectroscopy of adsorbed CO (DRIFTS CO) and X-ray photoelectron spectroscopy (XPS) techniques. The catalyst that was preliminarily reduced at 500 °C in $H_2$ was shown to contain PdIn intermetallic nanoparticles that demonstrate excellent diphenylethylene formation selectivity (~98%) in a 90% diphenylacetylene conversion. Oxidative treatment of the catalyst induces a preferential oxidative leaching of the In component from the intermetallic compound, modifying the Pd active site structure to a state with a high hydrogenation activity but poor stilbene selectivity. It was shown that the transformations occurring upon redox treatment are reversible and reproducible. The regularities of PdIn IMC surface post-synthesis modification via oxidative treatments under different conditions have been also investigated for model PdIn/HOPG samples [43,44]. It was shown that the reversible redox transformation $Pd-In_{IMC} \leftrightarrow Pd^0 + InO_x$ can be efficiently used to deliberately tune the nanoparticle surface composition/structure. Nevertheless, the restoration of the PdIn IMC phase always took place at high temperature ~500 °C, although the exact conditions and kinetics of the restoration of the intermetallic PdIn particle structure after their oxidation has remained unexplored so far. For example, Varga et al. [45] investigated rhodium nanoparticle size changes in the $Rh/CeO_2$ system during reduction by $H_2$ and heating in $O_2$ flow and in UHV as well. It was shown that, depending on metal loading and particle size, the temperature of full reduction of Rh can be varied with a further temperature increase, leading to sintering and agglomeration.

Thus, the current work is aimed at investigating the bulk and surface transformations of supported palladium–indium nanoparticles under oxidative treatments and the subsequent process of the restoration of the initial IMC structure after re-reduction of the $PdIn/Al_2O_3$ catalyst in hydrogen. Such information could help one gain insight into the reversible regeneration of the PdIn IMC during redox treatments. To achieve this goal, a set of techniques, viz., XRD, XPS and DRIFTS of adsorbed CO, were used in situ as appropriate methodology to gather the required structural and spectral information.

## 2. Materials and Methods

### 2.1. Sample Preparation

The bimetallic sample containing 2.5 wt.% Pd and 2.7 wt.% In was prepared via the incipient wetness impregnation of $Al_2O_3$ («Sasol», $S_{BET}$ = 56 $m^2/g$) with a solution of $PdIn(AcO)_5$ heterometallic acetate complex in dilute acetic acid with subsequent drying in air under ambient conditions. The resulting catalyst was reduced in 5% $H_2/Ar$ flow at 550 °C for 3 h and stored in a glass tube in an inert atmosphere.

The surface structure of the prepared catalyst was modified via the following set of sequential redox treatments:

(1)　Reduction in $H_2$ at 500 °C for 1 h;
(2)　Oxidation in $O_2$ at 300 °C for 1 h;
(3)　Reduction in $H_2$ at 50 °C for 1 h;
(4)　Reduction in $H_2$ at 100 °C for 1 h;
(5)　Reduction in $H_2$ at 150 °C for 1 h;
(6)　Reduction in $H_2$ at 200 °C for 1 h;
(7)　Reduction in $H_2$ at 250 °C for 1 h;
(8)　Reduction in $H_2$ at 300 °C for 1 h;
(9)　Reduction in $H_2$ at 350 °C for 1 h;
(10)　Reduction in $H_2$ at 400 °C for 1 h;
(11)　Reduction in $H_2$ at 450 °C for 1 h;
(12)　Reduction in $H_2$ at 500 °C for 1 h.

After each step, the catalyst was characterized via DRIFT spectroscopy of adsorbed CO and XPS spectroscopy.

### 2.2. X-ray Diffraction

The in situ XRD experiments were performed at the "Precision diffractometry II" beamline of the shared research center "Siberian Synchrotron and Terahertz Radiation Centre" (Novosibirsk, Russia) [46]. The working X-ray wavelength of 1.0102 Å was set through a single reflection from a perfect flat Si(220) crystal. The end-station diffractometer was equipped with an XRK-900 high-temperature flow reactor chamber (Anton Paar GmbH, Graz, Austria) and a position-sensitive parallax-free linear OD-3M detector (Budker Institute of Nuclear Physics SB RAS, Novosibirsk, Russia) [47]. The sample was treated as follows: (1) it was heated in a 10% $H_2$+Ar flow to 500 °C at a rate of 20 °C/min, kept for 30 min and then cooled down to RT at the same rate; (2) heated in a 20% $O_2$+Ar flow to 250 °C at a rate of 10 °C/min, kept for 30 min and then cooled down to RT at the same rate; (3) heated in a 10% $H_2$+Ar flow initially to 250 °C and then to 500 °C at a rate of 10 °C/min kept for 30 min at every temperature point, and then cooled down to RT at the same rate. The total gas flow was 100 sccm for all treatment steps. XRD patterns were measured continuously at a rate of 1 frame per min. The phase analysis was performed using the ICDD PDF-2 database [48]. The $\gamma$-$Al_2O_3$ support model was defined based on a pattern of a single-phase sample as a set of peaks and subtracted from the diffraction patterns before the catalyst profile parameters were refined. The crystallographic parameter refinement for the catalyst was performed using the MAUD v2.93 code [49].

### 2.3. Diffuse Reflectance Infrared Fourier-Transform Spectroscopy of Adsorbed CO

DRIFT spectra of adsorbed CO were obtained using a Tensor 27 spectrometer (Bruker Optics Inc., Billerica, MA, USA), equipped with an MCT detector and a high-temperature cell for in situ treatments (Harrick Scientific Products Inc., Pleasantville, NY, USA). Catalyst (a load of 20 mg) was milled and placed into the cell, then purged using an Ar flow for 10 min at 50 °C. After this pre-treatment, the background spectrum was recorded. Spectra of adsorbed CO for the «initial sample» were recorded at 50 °C under a 0.5% CO/He flow (30 $cm^3$/min) during 10 min. The next step was in situ reduction at 500 °C in a 5% $H_2$/Ar flow followed by cooling to 150 °C in the reductive atmosphere. Subsequent cooling to 50 °C was conducted under an Ar flow. Recording of the background spectrum and spectra of adsorbed CO was carried out according to the procedure described above. In the next steps, the sample was treated using a 20% $O_2$/$N_2$ (30 $cm^3$/min) flow at 25 °C and 250 °C for 30 min, followed by recording of the background spectrum and spectra of adsorbed CO at 50 °C after each step of the oxidative treatment. The final step was the reduction of the oxidized catalyst at 500 °C for 1 h in a 5% $H_2$/Ar flow.

The stepwise reduction of the Pd-In catalyst was performed after the oxidative treatment of the sample in a 20% $O_2$/$N_2$ flow at 350 °C for 1 h. Then, the cell was heated from 50 °C to 500 °C in a 5% $H_2$/Ar flow with an interval of 50 °C. The sample was kept for 30 min after reaching each temperature with subsequent cooling to 50 °C and recording of the spectra of adsorbed CO.

The registration of spectra was performed using the Opus 7.2.139.1294 software. After background correction, the collected spectra were deconvolved using the OriginPro software (9.0). For deconvolution of FTIR spectra, two Gaussian curves were used.

### 2.4. X-ray Photoelectron Spectroscopy

The XPS experiments were performed using a photoelectron spectrometer SPECS that consisted of four chambers: load lock, preparation, analyzer and high-pressure cell (HPC) chambers. The analyzer chamber is equipped with a PHOIBOS-150 hemispherical energy analyzer (SPECS, Berlin, Germany) and an Al$K_\alpha$ radiation source (h$\nu$ = 1486.6 eV, 200 W). The binding energy (BE) scale was pre-calibrated using positions of the Au$4f_{7/2}$ (BE = 84.0 eV) and Cu$2p_{3/2}$ (BE = 932.67 eV) core-level peaks. The residual gas pressure was better than $8 \times 10^{-9}$ mbar. For the measurements, the sample was supported on a stainless-steel mesh spot-welded onto a standard sample holder.

The HPC cell installed in one of the chambers of the spectrometer allows one to treat the sample using different gases at a pressure up to 1 bar over a temperature range from 50 °C to 500 °C. The temperature was controlled using a K-type thermocouple. After the pretreatment, the sample was cooled down to RT and the cell was pumped out to UHV conditions. Thus, the sample could be transferred to the analyzer chamber after pretreatment without contact with air.

The stepwise reduction of $PdIn/Al_2O_3$ catalyst was performed similarly to the afore-mentioned DRIFTS experiments. Firstly, the reduction of the initial sample (after prolonged storage in air) was performed in 50 mbar $H_2$ at 500 °C for 30 min. Then, the oxidative treatment of the sample in 200 mbar $O_2$ at 300 °C for 1 h was carried out. After that the sample was stepwise heated from 50 to 500 °C in 50 mbar $H_2$ with a step of 50 °C. The sample was kept for 30 min after reaching each temperature. The Al2p, C1s, O1s, Pd3d and In3d core-level spectra were measured after each treatment step.

Spectral analysis and data processing were performed with the XPS Peak 4.1 program [50]. Integrated line intensities were calculated from the area of the corresponding narrow regions (Al2p, C1s, O1s, Pd3d and In3d). For the quantitative analysis, the intensities of the corresponding photoelectron lines were corrected to respective atomic sensitivity factors [51].

## 3. Results and Discussion

### 3.1. X-ray Diffraction

XRD patterns of the $PdIn/Al_2O_3$ sample after being treated consecutively under different conditions (see Section 2) are presented in Figure 1. For the initial sample, no specific In- or Pd-containing phases were identified since all peaks were reliably recognized as those belonging to the $\gamma$-$Al_2O_3$ support. After the reduction step 1, the peaks (110), (200) and (211) of the PdIn 1:1 intermetallic compound emerged. The phase is characterized by the cubic structure of the CsCl type, space group Pm(-)3m and best-fit lattice parameter was a = 3.22(5) Å (a = 3.249 Å for PdIn [52]).

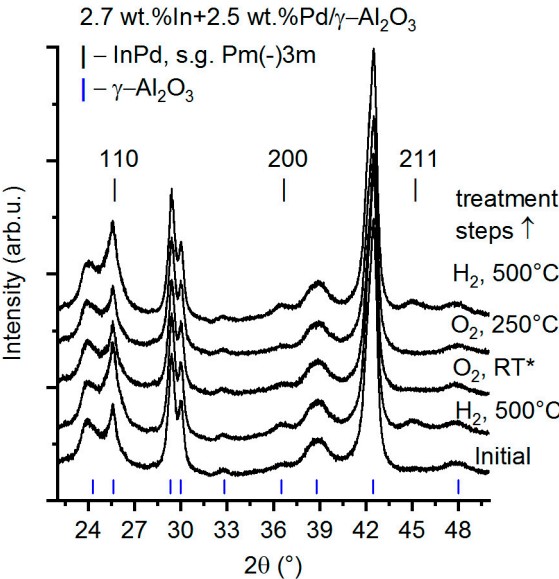

**Figure 1.** XRD patterns of $InPd/Al_2O_3$ after different treatments. *—reduced sample (step 1) after exposure to $O_2$ for 30 min at RT. Wavelength 1.0102 Å, all patterns were collected at RT.

The estimate of a crystallite size via the Scherrer formula [53] from the integral width of the 110 peak gives a value of 2.5–3 nm, which is close to the mean particle size calculated from TEM images obtained for the same catalysts in [42]. After the reduced sample was exposed to $O_2$ at room temperature (Figure 1, $O_2$ RT*), the reflexes of the PdIn intermetallic compound disappeared. Such a behavior can be explained by the In oxidation and segrega-

tion on the surface of the PdIn IMC particles, leading to a decrease in the crystallite size of the PdIn phase and respective smearing out of the diffraction peaks. The oxidation step 2 (20% $O_2$+Ar flow at 250 °C) induces no substantial changes in the phase composition. After the treatment, no In- or Pd-containing phases were identified, similar to the case of the initial catalyst (sample stored in air).

　　The formation of the intermetallic PdIn particles during subsequent treatment of the oxidized sample in a reductive atmosphere of 10% $H_2$+Ar (step 3) starts at 235 °C and continues up to 350 °C according to XRD data (Figure 2). Indices of the diffraction peaks belonging to the γ-$Al_2O_3$ support, which remain unchanged irrespective of treatment, are omitted for clarity. Lorenz et al. observed PdIn phase growth for an $In_2O_3$-supported Pd thin-film catalyst upon the hydrogen reduction at temperatures as low as 100 °C [52]. The lower temperature in that case is likely due to the thin-film nature of that catalyst. Iwasa et al. observed much higher temperatures of the $H_2$-TPR peak starting around 230 °C, with a maximum around 350 °C for an $In_2O_3$-supported Pd catalyst [54]. The PdIn phase was confirmed via XRD for the catalyst that reduced at 500 °C. The weight fraction of the PdIn phase was estimated from the intensity ratio between the γ-$Al_2O_3$ (440) peak located at 42.5° and PdIn (211) peak using the following set of parameters: γ-$Al_2O_3$ phase sp.gr. Fd(-)3m, a = 7.911 Å, theoretical density 3.657 g/cm$^3$ [55]; PdIn phase space group as above, theoretical density 10.7 g/cm$^3$. The crystallite size is shown to be virtually constant during heating in a reductive atmosphere up to 500 °C meaning that no considerable sintering of the particles occurs even after 30 min at 500 °C. The in situ XRD data are in good agreement with the aforementioned results obtained for the same PdIn/$Al_2O_3$ catalysts using XPS and DRIFTS CO techniques [42].

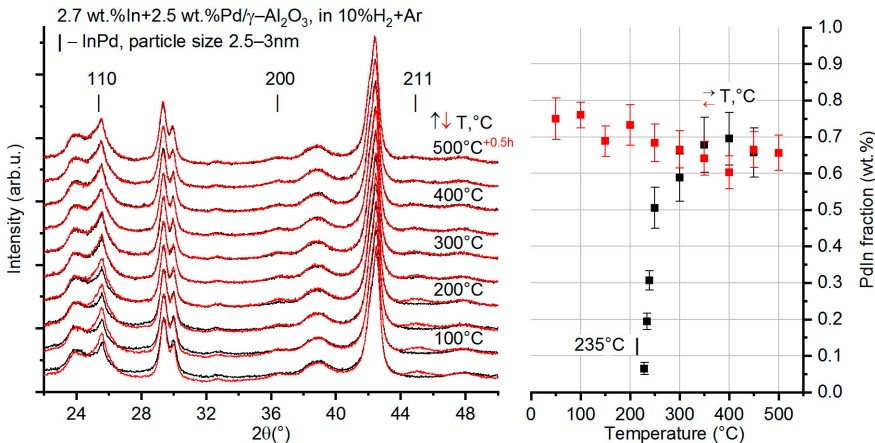

**Figure 2.** XRD patterns of InPd/$Al_2O_3$ during heating/cooling in 10% $H_2$+Ar at a rate of 10 °C/min (**left**) and corresponding changes in the phase composition (**right**). Wavelength 1.0102 Å. Black squares/lines correspond to the heating step and red squares/lines correspond to the cooling step.

### 3.2. DRIFT Spectroscopy of Adsorbed CO

　　The DRIFT spectrum of adsorbed CO for the initial sample reveals two broad asymmetric peaks in the region of 2200–1800 cm$^{-1}$ (Figure 3). The band with the maximum at 2086 cm$^{-1}$ corresponds to the linear form of CO adsorbed on palladium atoms, while a low-intensity peak in the range of 2000–1800 cm$^{-1}$ is assigned to the multi-bonded CO forms. According to [56], the position of the band related to linear CO is close to that in the spectra of monometallic palladium (~2080–2095 cm$^{-1}$). The appearance of peaks of bridge-bonded CO forms (~2000–1900 cm$^{-1}$) confirms the presence of monometallic palladium nanoparticles on the surface of the sample before in situ treatments.

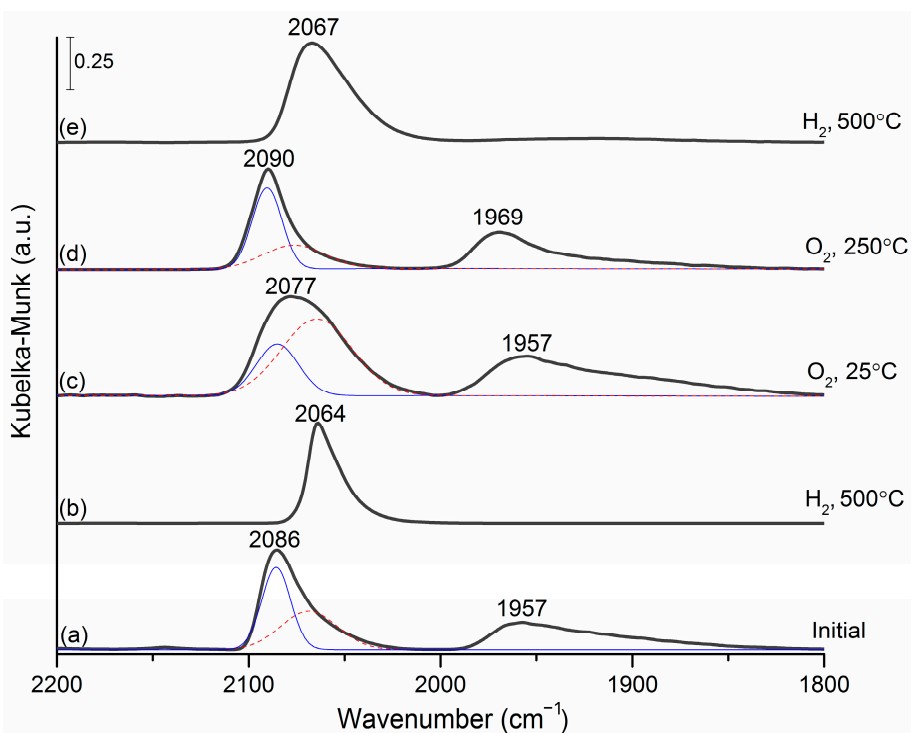

**Figure 3.** Normalized DRIFT spectra of adsorbed CO for the (**a**) initial sample; (**b**) catalyst after reduction at 500 °C in a 5% $H_2$/Ar flow 1 h; (**c**) oxidized at 25 °C for 30 min in an $O_2$/$N_2$ flow; (**d**) oxidized at 250 °C for 30 min; (**e**) reduced at 500 °C for 1 h.

The reduction of the catalyst under a flow of 5% $H_2$/Ar at high temperature leads to the disappearance of any absorption bands below 2000 $cm^{-1}$. This can be explained in terms of the formation of the intermetallic PdIn compound with palladium atoms effectively separated from each other by indium. The CO chemisorption in the bridged and hollow-bonded forms becomes energetically unfavorable in this case. In addition, the formation of bimetallic phase on the surface of the catalyst is accompanied by a shift in the peak of linearly adsorbed CO from 2086 to 2064 $cm^{-1}$. This shift may be related to a modification of palladium electronic states by Pd-In bonds [57–59]. Furthermore, the weakening of lateral interactions between adsorbed CO molecules could also lead to a shift towards low wavenumbers.

An oxidative treatment under ambient conditions gives rise to a broadening of the peak of linear CO and a reverse shift in its maximum from 2064 back to 2077 $cm^{-1}$. The appearance of a band, which corresponds to bridge-bonded CO, indicates the partial destruction of the intermetallic structure with the subsequent formation of «islands» of monometallic palladium. According to the deconvolution of the peak of linear CO, approximately 30 at.% of PdIn on the surface is converted from IMC to metallic Pd. The subsequent oxidation at 250 °C under a flow of 20% $O_2$/$N_2$ leads to the decomposition of ~60–65 at.% of the bimetallic phase. The peak of the linear CO form shifts from 2077 to 2090 $cm^{-1}$, which means significant changes in the catalyst surface structure.

The subsequent re-reduction of the sample in a $H_2$/Ar flow at 500 °C leads to the complete recovery of the intermetallic structure, as evidenced by the absence of absorption bands below 2000 $cm^{-1}$. These results are consistent with XRD measurements, according to which the complete restoration of the bulk structure of the intermetallic compound after oxidation can be achieved using a reductive treatment at 500 °C.

To search for a tighter correlation with in situ XRD data, a stepwise reduction of the PdIn/$Al_2O_3$ catalyst was carried out from 50 to 500 °C. Prior to the reductive treatments, the sample was oxidized at 350 °C for 30 min in an $O_2$/$N_2$ flow (Figure 4). The spectrum of the oxidized catalyst is characterized by an asymmetric peak with a maximum at

2090 cm$^{-1}$ and a broad band at 1980–1950 cm$^{-1}$. According to previous experiments, the appearance of these bands indicates the decomposition of the IMC phase with the formation of monometallic palladium particles [42]. The reduction of the sample at 50 °C for 30 min leads to a bathochromic shift in both bands with the emergence of a shoulder at 2080–2060 cm$^{-1}$. A further reductive treatment from 100 to 200 °C results in a shift in the maximum of the linear CO band to ~2060 cm$^{-1}$, with a gradual decrease in the intensity of the band, which corresponds to bridge-bonded CO. The disappearance of bands in the range of 2000–1800 cm$^{-1}$ at 200–300 °C may be related to the formation of the Pd-In IMC phase on the catalyst surface. A further increase in temperature leads to a small shift in the maximum of the linear band to 2064 cm$^{-1}$, confirming the formation of intermetallic PdIn nanoparticles. A slight shift from 2059 to 2064 cm$^{-1}$ observed with the temperature increase from 150–200 to ~350 °C may be associated with a small variation in surface the In/Pd atomic ratio during stepwise reduction.

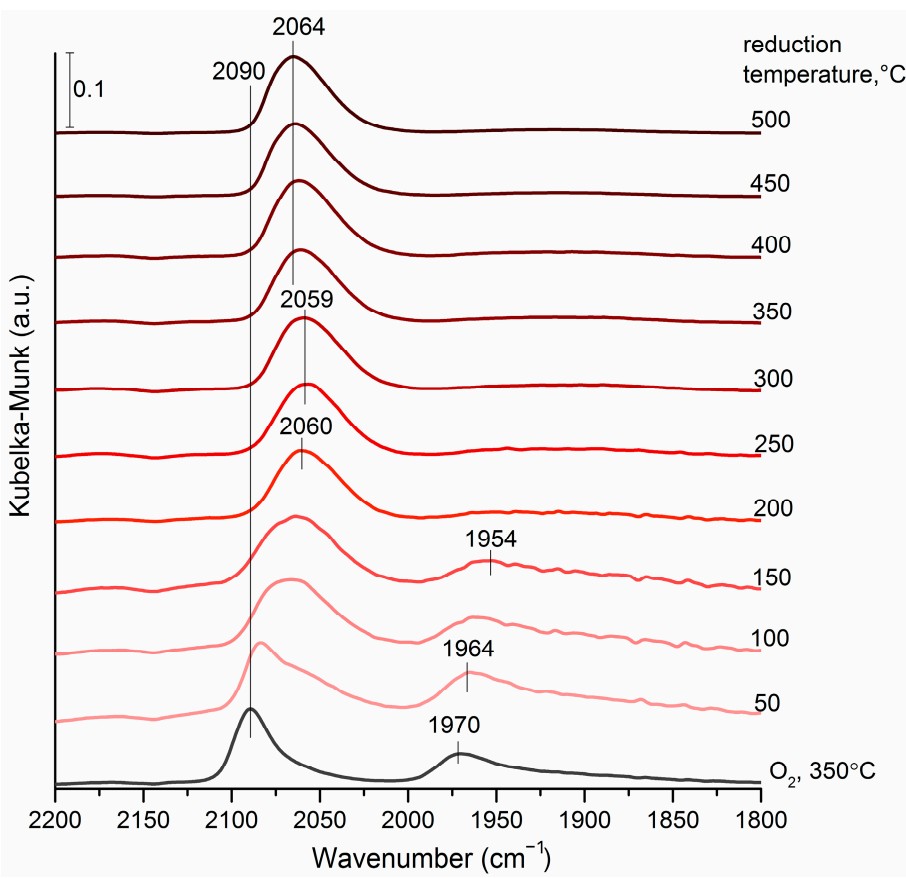

**Figure 4.** Normalized DRIFT spectra of adsorbed CO for the Pd-In/Al$_2$O$_3$ catalyst, reduced in 5% H$_2$/Ar at different temperatures. All spectra were recorded at 50 °C.

### 3.3. X-ray Photoelectron Spectroscopy

In order to study the transformations of PdIn nanoparticles oxidized at 300 °C upon reductive treatments at various temperatures, the surface composition and chemical states of the elements therein were thoroughly investigated via X-ray photoelectron spectroscopy.

Pd3d and In3d core-level spectra acquired for the sample that was first reduced in an atmosphere of 50 mbar H$_2$ at 500 °C and then re-oxidized at 200 mbar of O$_2$ at 300 °C are presented in Figure 5. The identification and deconvolution of the spectra (to retrieve such peak fitting parameters as peak positions, FWHM and relative intensities of different states) were carried out based on the results reported by us earlier [42]. As one can see, for the initial PdIn/Al$_2$O$_3$ catalyst (after prolonged storage in air), monometallic Pd (Pd$^0$), PdIn intermetallic species and PdO coexist on the surface, as indicated by the Pd3d$_{5/2}$ states at 335.1 eV, 335.8 eV and 337.2 eV, respectively [42–44,60–62]. As for indium, there

are two oxide states $In_2O_3$ and $InO_{surf}$ predominate in the In3d spectra at Bes ~445.3 eV and 444.2 eV, respectively [42–44,60–64]. A small fraction of $In_{IMC}$ is also present at a binding energy of 443.7 eV [42–44]. It should be mentioned that the In/Pd atomic surface ratio calculated from XPS data for the initial catalysts is 1.9, indicating that the surface is noticeably enriched with In species.

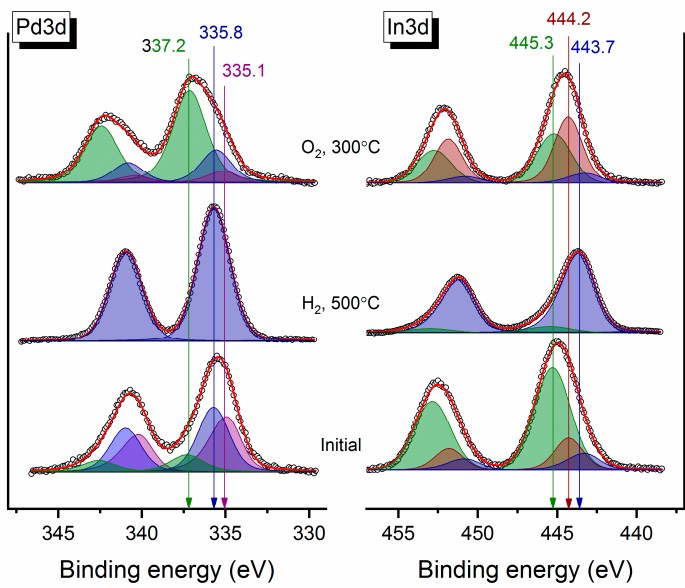

**Figure 5.** Pd3d (**left**) and In3d (**right**) spectra for $PdIn/Al_2O_3$ after different treatments.

The reduction of the initial catalyst in 50 mbar $H_2$ at 500 °C leads to the formation of the PdIn IMC phase. This is confirmed by the presence of only one $Pd3d_{5/2}$ peak with a binding energy of 335.8 eV, which is assigned to palladium atoms in IMC. At the same time, in In3d spectra, the $In3d_{5/2}$ component with BE of 443.7 eV ($In_{IMC}$) predominates; additionally, traces of $In_2O_3$ can be revealed in the In3d spectra. The In/Pd surface ratio dropped down to 1.2, which is actually close to the stoichiometry of the PdIn 1:1 intermetallic compound.

After oxidative treatment in $O_2$ at 300 °C, one can see a drastic change in the Pd3d and In3d XP spectra (Figure 5). Pd3d spectra reveal the appearance of PdO ($Pd3d_{5/2}$ = 337.2 eV) and metallic $Pd^0$ (335.1 eV), and the intensity of the $Pd_{IMC}$ (335.8 eV) state is noticeably decreased. Similar changes can be seen in the In3d spectra—the oxidized $InO_{surf}$ and $In_2O_3$ states (with $In3d_{5/2}$ BEs of 444.2 eV and 445.3 eV, respectively) predominate with just a small fraction of $In_{IMC}$. It should also be mentioned that the In/Pd atomic ratio is also increased up to 1.5, which points out the In segregation on the IMC particles' surface. Overall, the observed changes indicate the decomposition of a significant part of the intermetallic compound associated with the formation of palladium and indium oxides. Actually, this result is in good agreement with the results provided via XRD and DRIFTS techniques, as detailed above.

In the next step, the consecutive reduction of the oxidized $PdIn/Al_2O_3$ was performed in a temperature range from 50 to 500 °C (see Materials and Methods) in order to establish the exact conditions needed for the surface restoration of $PdIn_{IMC}$ particles.

Figure 6 shows the results of the deconvolution of corresponding Pd3d spectra and fractions of different Pd states, depending on the treatment conditions. Already after the reduction of the oxidized $PdIn/Al_2O_3$ catalyst in $H_2$ at 50 °C, the state assigned to PdO completely disappears from the XP spectra. Simultaneously, increases in the $Pd_{alloy}$ (from 22% to 68%) and $Pd^0$ (from 9% to 32%) fractions are observed. A further increase in reduction temperature up to 100 °C leads to a rise in the $Pd_{alloy}$ state fraction (~80%) and decrease in the $Pd^0$ fraction (~20%). Above 150 °C, only the $Pd_{alloy}$ state remains on the surface of the catalyst, indicating the full recovery of the PdIn IMC particles under these conditions.

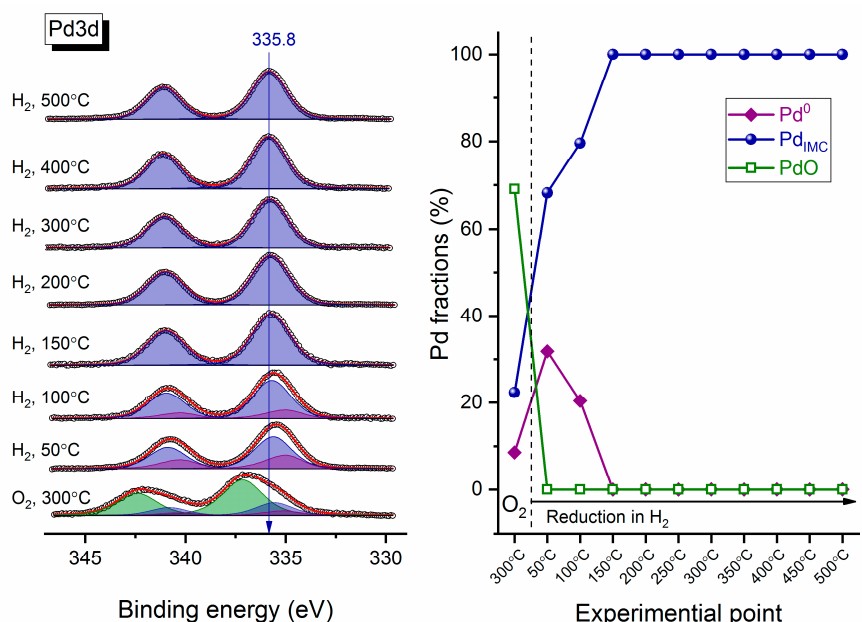

**Figure 6.** Results of the deconvolution of corresponding Pd3d spectra into individual Pd states and their fractions depending on the treatment conditions.

Figure 7 displays results of the deconvolution of corresponding In3d spectra and fractions of different In states, depending on the treatment conditions. After reduction of the oxidized PdIn/Al$_2$O$_3$ catalyst in H$_2$ at temperatures 50–150 °C, one can see a gradual increase in the In$_{IMC}$ fraction with a simultaneous decrease in the fraction of oxidized states (InO$_{surf}$ and In$_2$O$_3$). At a reduction temperature of 150 °C and above, the In$_{IMC}$ state predominates, and the spectra remain virtually unchanged up to 500 °C, also indicating the full recovery of the PdIn IMC particles already at 150 °C.

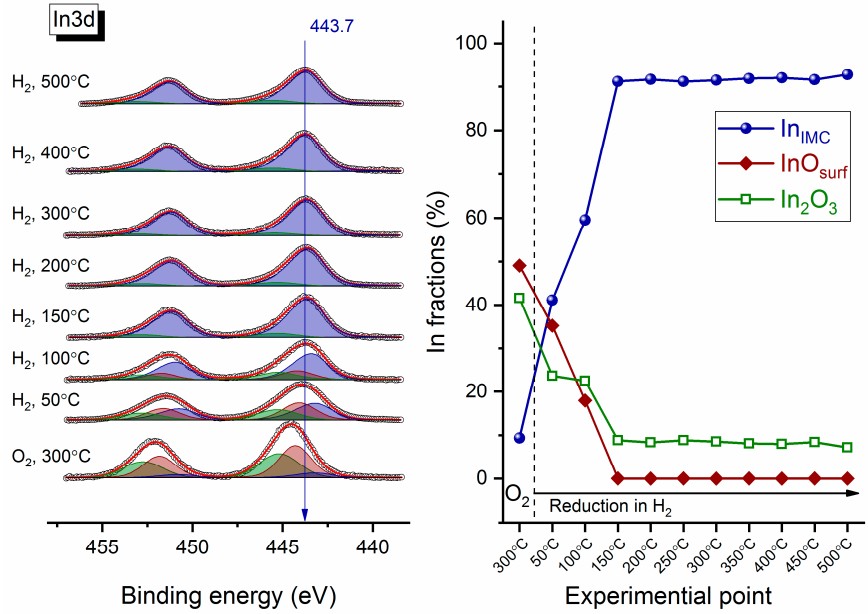

**Figure 7.** Results of the deconvolution of corresponding In3d spectra into different In states and their fractions depending on the treatment conditions.

The correlation of the atomic In/Pd and In$_{IMC}$/Pd$_{IMC}$ ratios for different reduction conditions is presented in Figure 8. As already mentioned, the In/Pd atomic ratio is 1.5 for the PdIn/Al$_2$O$_3$ catalyst oxidized at 300 °C, indicating In surface segregation. Meanwhile,

the $In_{IMC}/Pd_{IMC}$ atomic ratio is ~0.55, which indicates the presence of sub-stoichiometric IMC depleted with In. The reductive treatment at 50 °C leads to an In/Pd atomic surface ratio decrease down to ~1.35, together with a partial reduction of PdO and In oxides (see Figures 6 and 7), which points out a partial reverse redistribution of In and Pd atoms. At the same time, the $In_{IMC}/Pd_{IMC}$ atomic ratio increases up to 0.8, which, again, becomes quite close to the PdIn 1:1 stoichiometry. The following gradual increase in reduction temperature up to 150 °C leads to a further decrease in the In/Pd atomic ratio down to ~1.2 and an increase in the $In_{IMC}/Pd_{IMC}$ atomic ratio up to ~1.2, indicating the restoration of the PdIn IMC structure on the PdIn/Al$_2$O$_3$ surface. A further increase in the reduction temperature yields no noticeable change, either in In/Pd or in $In_{IMC}/Pd_{IMC}$ surface atomic ratios.

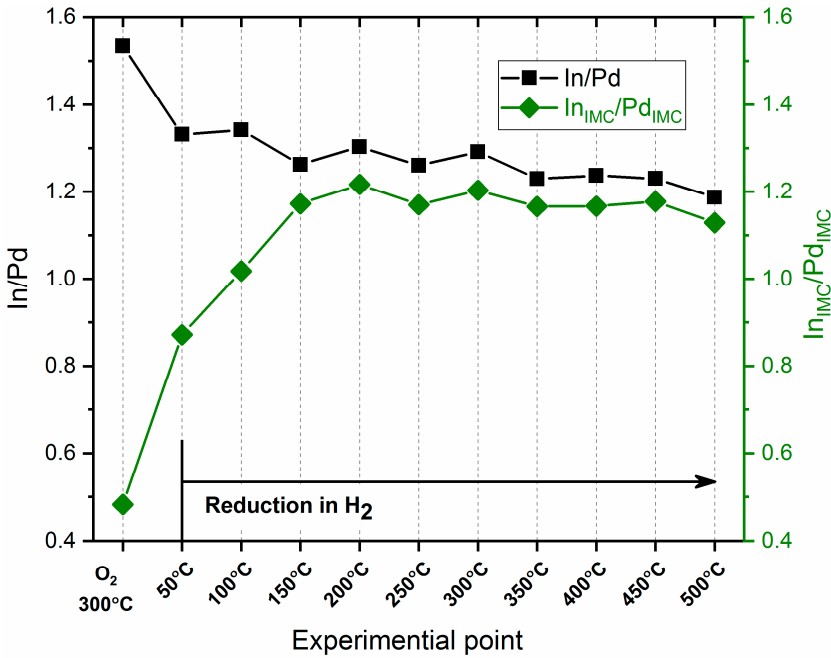

**Figure 8.** Correlation of atomic In/Pd and $In_{IMC}/Pd_{IMC}$ ratios depending on the reduction conditions.

The Pd3d$_{5/2}$ and In3d$_{5/2}$ BE values of the states corresponding to Pd and In in IMC as a function of the reduction temperature, as well as $In_{IMC}/Pd_{IMC}$ atomic ratio, are presented in Figure 9. For the PdIn/Al$_2$O$_3$ catalyst oxidized at 300 °C, the Pd3d$_{5/2}$ and In3d$_{5/2}$ core-level peaks are located at 335.5 eV and 443.2 eV, respectively. And, they gradually shift towards higher binding energy values for values of ~0.2 eV and ~0.4 eV, respectively, as reduction temperature increases from 50 to 150 °C. A further increase in the reduction temperature affords no noticeable shift in the Pd3d$_{5/2}$ and In3d$_{5/2}$ BE values. According to the literature, the shifts in the Pd3d$_{5/2}$ and In3d$_{5/2}$ BEs were associated with the formation of IMCs with different In/Pd atomic ratios [42,43,65–67]. A simple Pd$^{\delta+}$-In$^{\delta-}$ charge transfer model was suggested for a qualitative interpretation of such a behavior [42,43,60]. With increasing $In_{IMC}/Pd_{IMC}$ atomic ratio, a relative increase in the number of the In–In bonds (with the simultaneous decrease in the number of the In–Pd bonds) should cause a relative decrease in the local negative charge at In atoms, which, thus, should lead to a shift in the In3d peak to a higher binding energy. In turn, an increase in the Pd3d BE value is also expected because Pd should become more positively charged in the In-coordinated state relative to the "Pd only" state. Thus, the formation of a more "Pd only"-like state, i.e., contiguous Pd sites, could be suggested as a result of the oxidative treatment. The reduction of the oxidized sample at 150 °C leads to the full restoration of the initial IMC PdIn structure on the surface of the catalyst.

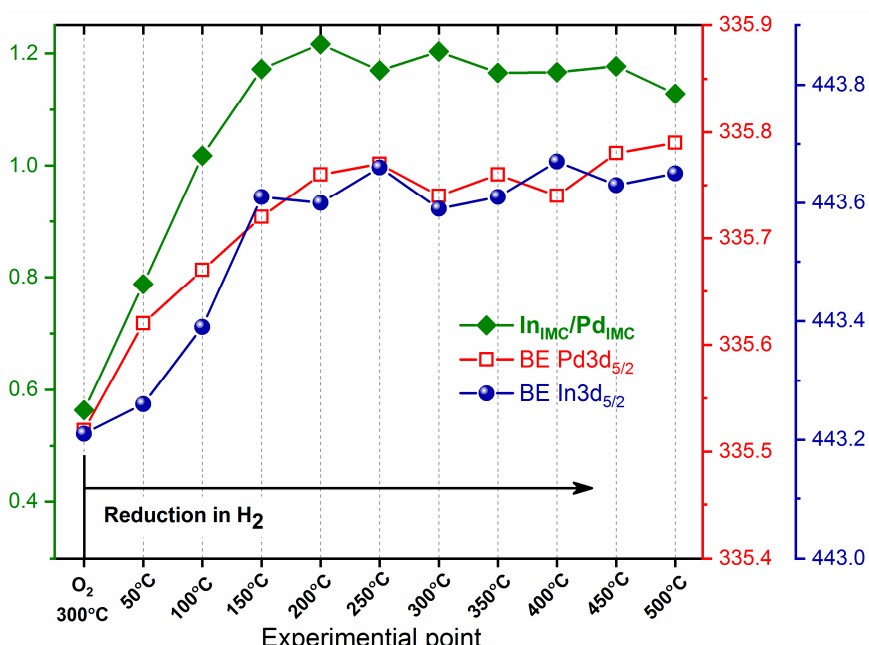

**Figure 9.** The Pd3d$_{5/2}$ and In3d$_{5/2}$ binding energies values of the states corresponding to Pd and In in IMC as a function of reduction temperature, as well as In$_{IMC}$/Pd$_{IMC}$ atomic ratio.

In summary, a detailed combined analysis of in situ XRD, XPS and DRIFTS CO allows us to discern several distinct stages of the IMC PdIn structure restoration for the oxidized PdIn nanoparticles upon their reduction at different temperatures, which are listed and described below.

(1) Reduction of the initial PdIn/Al$_2$O$_3$ catalyst (after prolonged storage in air) in H$_2$ at 500 °C leads to the formation of PdIn intermetallic nanoparticles. XPS data show the presence of only Pd$_{IMC}$ and domination of In$_{IMC}$ (with a minor admixture of In oxide) on the catalyst's surface. At the same time, the In$_{IMC}$/Pd$_{IMC}$ atomic ratio is 1.1, which is actually close to the nominal stoichiometry of the expected PdIn intermetallic phase. This suggestion is also in good agreement with DRIFTS CO data, which show the presence of only linearly adsorbed CO, characterized by a band at 2065 cm$^{-1}$, indicating that the CO adsorption occurs exclusively on single-atom Pd$_1$ surface sites completely isolated from each other by In atoms and, therefore, suggesting the formation of intermetallic nanoparticles with a regular PdIn structure [38–40]. Respective in situ XRD patterns also reveal broadened (110), (200) and (211) peaks of the PdIn 1:1 intermetallic nanoparticles.

(2) The subsequent oxidation of the reduced catalyst in O$_2$ at a temperature of 250–350 °C leads to the decomposition of a significant fraction of the PdIn intermetallic compound. According to XPS data, the fractions of Pd$_{IMC}$ and In$_{IMC}$ do not exceed 15–20%. At the same time, oxidized forms of Pd (PdO) and In (InO$_{surf}$ and In$_2$O$_3$) predominate on the surface. The In/Pd atomic ratio is increased up to ~1.5, indicating the segregation of oxidized In on the surface of intermetallic nanoparticles. At the same time, the In$_{IMC}$/Pd$_{IMC}$ atomic ratio drops down to 0.5, pointing out the depletion of the PdIn phase with In. DRIFTS CO data also indicate the decomposition of the IMC phase with the formation of monometallic palladium particles—the FTIR spectrum is characterized by an asymmetric peak with a maximum at 2090 cm$^{-1}$ and a broad band at 1980–1950 cm$^{-1}$. In situ XRD consistently identifies no In- or Pd-containing phases after the oxidative treatment, assuming the complete decomposition of the intermetallic particles to highly dispersed oxide forms, similar to the state of the initial catalyst (exposed to prolonged storage in air).

(3) Reduction of the oxidized PdIn/Al$_2$O$_3$ in H$_2$ at mild temperatures 50–100 °C: The reduction of the oxidized PdIn/Al$_2$O$_3$ catalyst in H$_2$ at 50 °C leads to the full reduction of the Pd component and a partial reduction of the In component with an increase in the PdIn IMC fraction. Indeed, XPS data show the complete disappearance of the PdO state in the Pd3d spectra and increase in the Pd$^0$ and Pd$_{IMC}$ fractions after the reduction of the sample at 50 °C. The In3d spectra show an increase in the In$_{IMC}$ fraction and a concerted decrease in the fractions of oxidized In species (InO$_{surf}$ and In$_2$O$_3$). The In/Pd atomic ratio drops down to ~1.35, indicating a reverse redistribution of the Pd and In components. An increase in the In$_{IMC}$/Pd$_{IMC}$ ratio points out the re-saturation of the PdIn IMC by In atoms. An increase in the reduction temperature up to 100 °C leads to a decrease in the Pd$^0$ fraction and an increase in the Pd$_{IMC}$ fraction, and a further reduction of oxidized In states occurs simultaneously. The presence of both Pd$^0$ and Pd$_{alloy}$ on the surface of the PdIn/Al$_2$O$_3$ catalyst after reduction at 50 and 100 °C points out the presence of different types of Pd sites—multi-atomic Pd$_n$ centers and single-site Pd$_1$ centers, respectively. It means that the ratio between those sites on the surface could be tuned by selecting a specific reduction temperature below 150 °C. Therefore, a novel strategy to deliberately tune the PdIn nanoparticle surface composition/structure could be suggested, in addition to the post-synthesis modification using an oxidative treatment, as described in [23]. This suggestion is in good agreement with the DRIFTS CO results. The broadening of the band at ~2060–2090 cm$^{-1}$ corresponds to the formation of both Pd site types (multi-atomic Pd$_n$ centers vs. single-site Pd$_1$ centers) on the surface of the PdIn IMC nanoparticles. At 100 °C, the shape of this broad band changes, indicating the redistribution of those two site types on the surface. According to XRD, no In- or Pd-containing phases could be identified. Presumably, the bulk structure of PdIn nanoparticles remains somewhat disordered after the reduction of the sample at such low temperatures (below 100 °C).

(4) Reduction of the oxidized PdIn/Al$_2$O$_3$ in H$_2$ at temperatures 150–500 °C: According to XPS data, the reduction at 150 °C leads to the full reduction of In oxides and also to the complete disappearance of Pd$^{0,}$ indicating the complete restoration of the PdIn IMC structure on the surface of nanoparticles. The Pd/In atomic ratio just slightly drops to ~1.2, indicating the further redistribution of Pd and In components. The In$_{IMC}$/Pd$_{IMC}$ ratio approaches ~1.2, pointing out the formation of the IMC with the stoichiometry close to the PdIn phase. A further increase in the reduction temperature affords no noticeable change in the XPS data. DRIFTS CO supports the mechanism of the nearly complete restoration of the PdIn IMC structure on the surface after the reduction of the PdIn/Al$_2$O$_3$ catalyst at a temperature above 150 °C. The complete disappearance of bands in a range of 2000–1800 cm$^{-1}$, which are associated with multiple CO bonds, at 200 °C point to the formation of the IMC PdIn phase on the catalyst surface. A further increase in temperature leads only to a small shift in the linear adsorption band maximum to 2064 cm$^{-1}$, confirming the formation of the intermetallic PdIn phase with the single-atom Pd$_1$ sites. According to in situ XRD data, the formation of the intermetallic PdIn particles upon a reduction treatment starts at 235 °C and continues up to 350 °C. Such a difference in temperature ranges needed for the restoration of the IMC PdIn structure, as retrieved through surface-sensitive (XPS and DRIFTS CO) and bulk (XRD) techniques allow us to suggest that the recovery of the IMC structure on the surface of PdIn nanoparticles occurs at a lower temperature of ~150 °C; the complete transformation into regularly ordered bulk PdIn phase requires temperatures as high as ~350 °C.

## 4. Conclusions

The combined utilization of bulk- (XRD) and surface-sensitive (XPS and DRIFTS CO) techniques makes it possible to reconstruct a comprehensive picture of processes occurring with the PdIn/Al$_2$O$_3$ catalyst upon reduction from the initially oxidized state. The recovery of the chemical states of Pd and In atoms typical of the PdIn IMC on the

surface nanoparticles occurs at relatively low temperature ~150 °C, whilst the formation of a long-range-ordered bulk PdIn phase requires a substantially higher temperature of ~350 °C. Therefore, a «novel» strategy to deliberately tune the PdIn nanoparticle surface composition/structure could be suggested for the optimization of their catalytic properties. An appropriate selection of the reduction temperature applied to the oxidized PdIn/Al$_2$O$_3$ catalyst in a range below 150 °C could provide a desired balance between the two distinctly different types of the Pd sites—multi-atomic Pd$_n$ centers and single-site Pd$_1$ centers on the catalyst's surface. The regular and most ordered structure of the PdIn intermetallic compound violated via oxidation can be recovered in a straightforward and fully reversible way through a reductive treatment at a high temperature (~500 °C).

**Author Contributions:** Conceptualization, A.V.B., Y.V.Z., V.I.B. and A.Y.S.; methodology, A.V.B., M.A.P., I.P.P., N.S.S., Z.S.V. and A.Y.S.; software, A.V.B., M.A.P., N.S.S. and Z.S.V.; validation, A.V.B., I.P.P. and Y.V.Z.; formal analysis, A.V.B., I.P.P., I.S.M., G.N.B., G.O.B. and P.V.M.; investigation, A.V.B., M.A.P., I.P.P., N.S.S., Z.S.V., I.S.M., G.O.B., P.V.M. and G.N.B.; data curation, A.V.B., M.A.P., Z.S.V., N.S.S., Y.V.Z., V.I.B. and A.Y.S.; writing—original draft preparation, A.V.B., M.A.P., N.S.S. and Z.S.V.; writing—review and editing, I.P.P., Y.V.Z., V.I.B. and A.Y.S.; visualization, A.V.B., N.S.S. and Z.S.V.; supervision, A.V.B., Y.V.Z., V.I.B. and A.Y.S.; project administration, V.I.B. All authors have read and agreed to the published version of the manuscript.

**Funding:** This work was supported by the Ministry of Science and Higher Education of the Russian Federation (Agreement No. 075-15-2022-263).

**Data Availability Statement:** Not applicable.

**Acknowledgments:** The investigations were performed using large-scale research facilities "EXAFS spectroscopy beamline" at the Siberian synchrotron and terahertz radiation center.

**Conflicts of Interest:** The authors declare no conflict of interest.

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
