# Peer review of "Reversible Transformations of Palladium–Indium Intermetallic Nanoparticles upon Repetitive Redox Treatments in H2/O2"

_crystals, doi:10.3390/cryst13091356_

Round 1
Reviewer 1 Report
crystals-2573192
Reversible Transformations of Palladium-Indium Intermetallic Nanoparticles upon Repetitive Oxidative-Reductive Treatments
Andrey V. Bukhtiyarov, Maxim A. Panafidin, Igor P. Prosvirin, Nadezhda S. Smirnova, Pavel V. Markov, Galina N. Baeva, Igor S. Mashkovsky, Galina O. Bragina, Zakhar S. Vinokurov, Yan V. Zubavichus, Valerii I. Bukhtiyarov and Alexander Yu. Stakheev
The authors study the transformations of chemical states and structures in a PdIn/Al2O3 catalyst under different conditions. They found that under fully controlled conditions they can succeed any desired state of the catalyst between two opposite extremes of highly dispersed oxide species and regularly ordered Pd1In1 intermetallic compound. This is important for the catalytic performance since the catalyst speciation can be influence the catalytic activity and selectivity
The work is good, and convincing. The manuscript can be accepted after revision.
XRD patterns: What are the other main peaks of the catalyst? Please add the information in text
Why the peak of (211) has lower intensity with the reduction temperature? Please comment
Lines 238-242 The re-reduction is not fully reversible. Please compare the two peaks at 2067 and 2064 cm-1 for the spectra (e) and (b). The width of these two peaks is completely different
Figure 4 the values 2059,2060 and 2064 cm-1are significant different or in the experimental error? If not I suggest the author to erase these values from the figure
Please change the abstract in the part where hydrogenation of multiple C-C was referred.
Please check the keywords and try to simplified in order to gain visibility
English are fine
Author Response
1) XRD patterns: What are the other main peaks of the catalyst? Please add the information in text
We thank the reviewer for pointing this out. Actually, other peaks, which are not indexed in Figures 1 and 2 belong to the nanocrystalline γ-Al2O3 support and remain unchanged irrespective of the redox treatments. They are deliberately left unmarked for clarity. Taking into account the reviewer’s comment we have modified the Figure 1 by adding blue ticks explicitly indicating expected positions of these peaks originated from the γ-Al2O3 support. In order not to clutter up the Figure 2 we haven't added indices of peaks belonging to the γ-Al2O3 support. But we have tried to improve on the clarity and readability of the manuscript by correcting the corresponding sentences as follows:
For the initial sample, no specific In or Pd-containing phases were identified apart from the support peaks.
Indices of the diffraction peaks belonging to the γ-Al2O3 support, which remain unchanged irrespective of treatment, are omitted for clarity.
2) Why the peak of (211) has lower intensity with the reduction temperature? Please comment
We thank the reviewer for mentioning this. Actually, there are no statistically significant changes in the intensity of the (211) peak for the reduced sample during cooling from 500°C to RT. The apparent variations are possibly due to the thermal shift of the peak position towards higher diffraction angles and the comparative nature of the data presentation in Figure 2.
3) Lines 238-242. The re-reduction is not fully reversible. Please compare the two peaks at 2067 and 2064 cm-1 for the spectra (e) and (b). The width of these two peaks is completely different
We thank the reviewer for pointing this out. The difference of 3 cm-1 between 2064 and 2067 cm-1 is within the experimental error. Some width difference can be related to slight palladium surface inhomogeneity after reduction treatment (PdIn compound can exist in a wide range of Pd concentrations, from 45 to 61.5 at.% Pd [H. Okamoto, J. Phase Eq. 24 (2003) 481]), but the electronic and phase states of PdIn after oxidative-reductive treatments, according to DRIFTS CO and XRD results, are essentially the same.
The following sentence has been added to the revised manuscript:
A slight shift from 2059 to 2064 cm-1 observed with the temperature increase from 150-200 to ~ 350 °C may be associated with small variation of surface In/Pd atomic ratio during stepwise reduction.
4) Figure 4 the values 2059,2060 and 2064 cm-1are significant different or in the experimental error? If not I suggest the author to erase these values from the figure
We thank the reviewer for pointing this out. The experimental error in DRIFTS-CO technique is ca. 4 cm-1. The values 2060 and 2064 cm-1 are thus within the experimental error, but 2059 cm-1 is already beyond the experimental error. The origin of this scatter is unclear. It may be associated with a partial phase transformation of the Pd-In system or variation of surface Pd/In atomic ratio during the stepwise reduction.
5) Please change the abstract in the part where hydrogenation of multiple C-C was referred.
We thank the reviewer for pointing this out. We have modified the abstract of the revised manuscript by removing this information.
6) Please check the keywords and try to simplified in order to gain visibility
We thank the reviewer for pointing this out. We have simplified the keywords to gain their visibility.
Reviewer 2 Report
The PdIn/Al2O3 catalyst upon redox treatments in different gaseous atmospheres at different temperatures was investigated by different bulk and surface sensitive techniques. The manuscript contains valuable information and after minor correction and supplementation is suitable for publication. See the remarks below.
1. Recently the behavior and importance of metallic particles in material-, surface science and catalysis was revived in a nice Review (Catalysis Letters 151, (2021) 2153-2175. https://doi.org/10.1007/s10562-020-03477-5. Please refer to it in the Introduction.
2. Pd-Au (F. Gao, Y. Wang, D.W. Goodman J. Phys. Chem. C, 114 (2010), p. 4036) and Rh-Au system were widely investigated. Alloy formation and core shell structure were observed. Perhaps, the Introduction part would be supplemented with these systems. The related question is: Can you exclude the core-shell formation in the case of palladium-indium intermetallic system taken into account the differences in surface free enthalpy?
3. It is a nice result that repetitive oxidative-reductive treatment in different temperatures resulted in transformation of chemical state and structures in the Pd-In system. Similar experiments were carried out on Rh/CeO2 system. Temperature –induced agglomeration and redispersion upon reoxidation were established due to Rh-O-Ce bond (E. Varga et al. Langmuir, 32 (2016) 2761-2770, https://doi.org/10.1021/acs.langmuir.5b04482). Please consider these findings in the present discussion.
Author Response
- Recently the behavior and importance of metallic particles in material-, surface science and catalysis was revived in a nice Review (Catalysis Letters 151, (2021) 2153-2175. https://doi.org/10.1007/s10562-020-03477-5. Please refer to it in the Introduction.
We thank the reviewer for pointing this out. We have added this reference to the Introduction of the revised manuscript (please see reference 8).
- Pd-Au (F. Gao, Y. Wang, D.W. Goodman J. Phys. Chem. C, 114 (2010), p. 4036) and Rh-Au system were widely investigated. Alloy formation and core shell structure were observed. Perhaps, the Introduction part would be supplemented with these systems. The related question is: Can you exclude the core-shell formation in the case of palladium-indium intermetallic system taken into account the differences in surface free enthalpy?
We thank the reviewer for pointing this out. Actually, substitutional solid solutions, such as Pd-Au, Pd-Ag Pd-Cu, and others, can form either uniformly alloyed nanoparticles or nanoparticles with a “core-shell” structure depending on the preparation procedure or reaction conditions. Such systems are characterized by a high mobility of atoms inside the nanoparticles. Due to this fact, the surface of bimetallic nanoparticles based on substitutional solid solution can be easily tuned by a CO-induced segregation or under exposure to a reaction mixture. Another type of bimetallic system is represented by intermetallic compounds (for Pd-In, Pd-Ga and other systems), which are to the contrary characterized by a high degree of atomic order and crystal structure stability. Bimetallic Pd-In nanoparticles can form different stable intermetallic compounds (e.g., InPd, In3Pd2, In7Pd3) depending on the metal atomic ratio, where indium and palladium have well-ordered pattern of alternation. Since such systems are of higher stability, it is not so easy to destroy such structures, for example, by CO exposure as it is the case for substitutional solid solutions. To achive this goal, harsher conditions (for example, O2 treatment) are required. Concerning the probably occurrence of core-shell structures in Pd-In bimetallic nanoparticles, it was shown in our previously published papers devoted to bimetallic Pd-M/HOPG (M=Au, Ag, In) that a consecutive deposition (in the cases of Pd-Au and Pd-Ag) leads to the formation of bimetallic particles with a “core-shell” structure on the HOPG surface. To afford a regular alloy, some further annealing treatment under UHV conditions at 400–500 °C was necessary [A.V. Bukhtiyarov, I.P. Prosvirin, V.I. Bukhtiyarov, XPS/STM study of model bimetallic Pd-Au/HOPG catalysts, Appl. Surf. Sci. 367 (2016) 214–221; M.A. Panafidin, A.V. Bukhtiyarov, I.P. Prosvirin, I.A. Chetyrin, V.I. Bukhtiyarov, Model bimetallic Pd–Ag/HOPG catalysts: An XPS and STM study, Kinet. Catal. 59 (2018) 776–785.]. At the same time, the formation of Pd-In surface alloy occurs spontaneously and already at the step of In deposition on the surface of monometallic Pd/HOPG sample instead of formation of nanoparticles with a core-shell structure [please see Refs. 26, 27 of the revised manuscript].
- It is a nice result that repetitive oxidative-reductive treatment in different temperatures resulted in transformation of chemical state and structures in the Pd-In system. Similar experiments were carried out on Rh/CeO2 system. Temperature –induced agglomeration and redispersion upon reoxidation were established due to Rh-O-Ce bond (E. Varga et al. Langmuir, 32 (2016) 2761-2770, https://doi.org/10.1021/acs.langmuir.5b04482). Please consider these findings in the present discussion.
We thank the reviewer for figuring this out. We have extended the Introduction by adding the following information:
For example, Varga et al. [26] investigated rhodium nanoparticles size changes in Rh/CeO2 system during reduction by H2 and heating in O2 flow and in UHV as well. It was shown that depending on metal loading and particle size temperature of full reduction of Rh can be varied with further temperature increase leading to sintering and agglomeration.
Round 2
Reviewer 1 Report
The authors revised their manuscript according to my suggestions. The manuscript can be accepted